# A_2A_R as a Prognostic Marker and a Potential Immunotherapy Target in Human Glioma

**DOI:** 10.3390/ijms24076688

**Published:** 2023-04-03

**Authors:** Soumaya Rafii, Amina Ghouzlani, Oumayma Naji, Saadia Ait Ssi, Sarah Kandoussi, Abdelhakim Lakhdar, Abdallah Badou

**Affiliations:** 1Immuno-Genetics and Human Pathologies Laboratory, Faculty of Medicine and Pharmacy, Hassan II University, Casablanca 20000, Morocco; 2Department of Neurosurgery, UHC Ibn Rochd, Casablanca 20250, Morocco; 3Mohammed VI Center for Research and Innovation, Rabat, Morocco and Mohammed VI University of Sciences and Health, Casablanca 82403, Morocco

**Keywords:** human glioma, immune system, immune checkpoints, *A_2A_R*, transcriptomic level

## Abstract

Gliomas are considered one of the most malignant tumors in the body. The immune system has the ability to control the initiation and development of tumors, including gliomas. Thus, immune cells find themselves controlled by various molecular pathways, inhibiting their activation, such as the immunosuppressive adenosine 2A receptor (A_2A_R). Our objective was to establish the expression profile and role of *A_2A_R* at the transcriptomic level, using real-time RT-PCR in Moroccan glioma patients, in addition to TCGA and CGGA cohorts. The real-time RT-PCR results in Moroccan patients showed that high expression of this gene was associated with poor survival in males. Our study on the CGGA cohort corroborated these results. In addition, there was a positive association of *A_2A_R* with T-cell exhaustion genes. *A_2A_R* also correlated strongly with genes that are primarily enriched in focal adhesion and extracellular matrix interactions, inducing epithelial mesenchymal transition, angiogenesis, and glioma growth. However, in the TCGA cohort, the *A_2A_R* showed results that were different from the two previously examined cohorts. In fact, this gene was instead linked to a good prognosis in patients with the astrocytoma histological type. The correlation and enrichment results reinforced the prognostic role of *A_2A_R* in this TCGA cohort, in which its high expression was shown to be related to lymphocyte differentiation and a successful cytolytic response, suggesting a more efficient anti-tumor immune response. Correlations and differential analyses based on *A_2A_R* gene expression, to understand the cause of the association of this gene with two different prognoses (CGGA males and TCGA Astrocytoma), showed that the overexpression of A_2A_R in Chinese male patients could be associated with the overexpression of extracellular adenosine, which binds to A_2A_R to induce immunosuppression and consequently a poor prognosis. However, in the second group (TCGA astrocytomas), the overexpression of the gene could be associated with an adenosine deficiency, and therefore this receptor does not undergo activation. The absence of *A_2A_R* activation in these patients may have protected them from immunosuppression, which could reflect the good prognosis. *A_2A_R* can be considered a promising therapeutic target in male CGGA and Moroccan patients with gliomas.

## 1. Introduction

Gliomas are the most aggressive brain tumors [1,2,3]. They represent a major unsolved clinical problem. Thus, despite the ability of conventional therapies to prolong patient survival, most gliomas limit life expectancy. Therefore, there is an ongoing effort to explore potential immunotherapy strategies, as a solution to this major unsolved clinical problem.

In addition to the defense against various exogenous risks that endanger the human body, the immune system plays a crucial role in the maintenance of intrinsic homeostasis, through various immune checkpoints [4]. Unfortunately, these checkpoints can be manipulated by tumors, to evade immune surveillance and destruction. This means that the immune cells responsible for destroying cancer cells are controlled by these checkpoints, which inhibits their activation and effectiveness [4]. Unlike most strategies used to treat cancer, immune checkpoint therapy does not directly target tumor cells, but rather acts on immune cells, to stimulate their anti-tumor activity [5]. It has been shown that the majority of glioma patients do not respond to blockade of the usual immune checkpoint pathways, such as programmed death receptor-1 (PD1), its ligand (PD-L1), and cytotoxic T-lymphocyte antigen 4 (CTLA-4) [6,7,8]. This sparked our interest in searching for new immune checkpoints, whose targeting could bring new hope to glioma patients.

Among the immunosuppressive checkpoints coexisting in the tumor microenvironment, immunosuppressive adenosine receptors have been the subject of several studies on important immunosuppressive effects in various cancers [9,10]. Two adenosine receptors, the adenosine 2A receptor (A_2A_R) and the adenosine 3A receptor (A_3A_R), have been reported to be overexpressed in gliomas [11,12].

Furthermore, the A_2A_R has a strong affinity for adenosine [13]. We directed our research towards the study of the A_2A_R in gliomas, it is an inhibitory checkpoint of the immune system, which by blocking it with an antagonist has shown efficacy in other types of cancer, by improving the anti-tumor immune response [14,15,16,17,18]. This receptor is highly expressed in lymphocytes, macrophages, mast cells, neutrophils, dendritic cells, eosinophils, NK cells, endothelial cells, and epithelial cells [19]. Elevated *A_2A_R* expression was observed in glioma samples compared to healthy and peri-tumor tissues, with higher expression in grade III astrocytomas [12]. Therefore, blocking A_2A_R inhibited glioma proliferation [20]. Furthermore, a study showed that the A_2A_R pathway is highly expressed in gliomas on TCD8 cells, followed by the PD-1 pathway [21]. This prompted us to perform further investigations of the expression of *A_2A_R* at the transcriptomic level and of the immunosuppressive role mediated by this receptor in the tumor microenvironment of human gliomas.

## 2. Results

### 2.1. Evaluation of A_2A_R Expression According to Clinico-Pathological Parameters in a Moroccan Glioma Patient Cohort

Real-time RT-PCR was used to identify glioma samples that overexpressed the *A_2A_R* gene, and its expression was evaluated relative to clinical parameters (Table 1). A statistical analysis, using an unpaired *t*-test of the median between two patient groups, revealed no significant variation in the *A_2A_R* expression level.

To determine the potential relationship between *A_2A_R* expression and prognosis, patients were divided into two groups based on their level of A_2A_R expression. The first group had low median expression and the second group had high median expression. These two groups were analyzed for each clinical parameter, which allowed us to compare the two survival curves established using the Kaplan–Meier method. For each patient, we had a duration of observation per month and the status of the observation (Deceased/Living). The results revealed a highly significant association of A_2A_R expression with prognosis, only in men; the higher the gene expression, the worse the survival in males (*p* value = 0.0062) (Table 2). The survival curve plot of *A_2A_R* expression in Moroccan males is shown in Figure 1.

### 2.2. Confirmation of the Correlation between A_2A_R Gene Expression and Prognosis Using the Chinese Glioma Database (CGGA)

We conducted a study to examine the impact of *A_2A_R* expression on the survival of patients in the CGGA cohort (Table 3). The results showed that poor survival was associated with two subgroups: patients above the median age (44 years) and males. To ensure that the prognosis related to *A_2A_R* expression was influenced only by one of these parameters, we used multi-variate COX regression analysis. For this purpose, patients were divided into two groups, the first containing 109 patients with low *A_2A_R* expression (Figure 2a) and the second containing 106 patients with high *A_2A_R* expression (Figure 2b). The prognosis according to age did not change in the case of high *A_2A_R* expression compared to low *A_2A_R* expression; however, for the parameter gender, males presented a poor prognosis only in the case of high *A_2A_R* expression [hazard ratio (HR) = 1.71; *p* = 0.038]. These results are in agreement with those observed in Moroccan patients, where overexpression of *A_2A_R* in males was associated with poor survival. The plot of survival versus A_2A_R expression in CGGA males is shown in Figure 3.

### 2.3. A_2A_R Expression Is Most Elevated Compared to Other Immune Checkpointsin Male CGGA

Using the CGGA male dataset, the expression profile of *A_2A_R* was then compared to the expression of three critical immune checkpoints (PD-1, PD-L1, CTLA-4) known to be highly expressed in gliomas. The results showed that the *A_2A_R* expression level appeared to be elevated in the CGGA male tumor microenvironment (*p* < 0.0001) (Figure 4).

### 2.4. Identification of Molecular Profile Related to A_2A_R Expression in Male Glioma Patients (CGGA)

We aimed to identify a clear molecular profile of the microenvironment that predominantly surrounds gliomas in males in association with *A_2A_R* expression. We established a differential analysis based on *A_2A_R* expression. We obtained a list of 114 genes positively strongly correlated with *A_2A_R* (*p* value ≤ 0.05 and log2 FC ≥ 0.5). Next, we explored the biofunction of these genes through KEGG and hallmark enrichment analysis. The genes positively correlating with *A_2A_R* were mainly enriched in focal adhesion, extracellular matrix interaction and cell cycle, epithelial mesenchymal transition, and angiogenesis, which are related to glioma development (Figure 5) (Table 4). These findings suggested that *A_2A_R* may affect the formation of focal adhesions in gliomas and regulate the presence of ECM genes that are responsible for glioma development. These results further supported the role of *A_2A_R* in the development and aggressiveness of gliomas.

### 2.5. Correlation between A_2A_R Gene Expression and Prognosis of TCGA Glioma Patients

Next, we sought to compare the previous results with the TCGA cohort. Our findings in the TCGA cohort were contrasting, as they diverged from the results obtained from the previous two cohorts; we found that *A_2A_R* expression was related to a good prognosis, and this was observed in patients with an astrocytoma (Table 5).This is depicted in the survival curve plot of *A_2A_R* expression in TCGA patients with astrocytoma histological type, which is shown in Figure 6.

### 2.6. Identification of Pathways Related to A_2A_R Expression in Astrocytoma Glioma Patients (TCGA)

To determine the pathways related to the expression of *A_2A_R* in patients with astrocytoma, we performed a differential analysis followed by KEGG enrichment. We identified 576 genes that were positively correlated with *A_2A_R* expression. These genes were mainly enriched in myogenesis (Table 6), the Notch pathway, and Th1 and Th2 differentiation (Figure 7). This was in agreement with the good prognosis of *A_2A_R* in these patients and suggests a lack of immunosuppression associated with a high level of this gene.

### 2.7. Relation between A_2A_R Expression and T Cell Exhaustion in Male CGGA and TCGA Astrocytoma Patients

To investigate the relationship between *A_2A_R* expression and the state of the immune system, we compared this gene with the expression of T cell exhaustion genes (GET) in the two previously studied cohorts (male CGGA and TCGA astrocytomas). The results showed that in the male CGGA patients, there was a positive and significant association between A_2A_R expression and six T cell exhaustion genes, which were *LAG-3*, *CTLA-4*, *CXCR6*, *CD276*, *NKG7*, and *HLA-DRB1* (Figure 8a). However, in the TCGA astrocytoma cohort, there was a negative association of *A2AR* with *PD-L1*, *TIM-3*, and *HLA-DQA1*, and only a positive correlation with *LAG-3* (Figure 8b). This was subsequently reinforced by correlations of TCD8 infiltration levels with cytolytic activity score (Figure 9). We quantified the level of immune cell infiltration using Cibersortx and calculated the cytolytic activity score in patients with high A_2A_R expression, and then correlated TCD8 lymphocytes with cytolytic activity score. The results showed, in TCGA astrocytoma patients, a positive correlation between TCD8 (r = 0.16; *p* = 0.045) with the cytolytic activity score, suggesting an activated state of the anti-tumor immune response in these patients, whereas, in male CGGA patients, there was no correlation of TCD8 with cytolytic activity score.

### 2.8. Evaluation of A_2A_R Expression with Genes Responsible for Extracellular Adenosine Levels (CGGA Males and TCGA Astrocytoma)

To understand the underlying causes of the association of *A_2A_R* with the two different prognoses, in the two cohorts (CGGA males and TCGA Astrocytoma), we performed a differential analysis based on the A_2A_R gene expression in each group. We found that 102 genes were overexpressed in the presence of the *A_2A_R* gene in male CGGA patients (Figure 10a) and 306 genes were overexpressed in TCGA astrocytoma patients (Figure 10b). Genes that were overexpressed in both groups were excluded (11 genes) (Figure 10c). Afterward, we performed INOH-path enrichment, to determine the different elements and pathways that were uniquely overexpressed in association with *A_2A_R* in the microenvironment of CCGA males compared to the TCGA astrocytoma group. The results obtained showed an overexpression of genes that regulate pathways of pyruvate kinase deficiency, glutaminolysis, hyperphenylalanimia, and sugar metabolism in the group whose *A_2A_R* was associated with a poor prognosis (CGGA men) compared to the group whose gene was associated with a good prognosis (TCGA astrocytoma) (Table 7).

Furthermore, as the immunosuppressive effect of *A_2A_R* is dependent on the amount of extracellular adenosine, we correlated *A_2A_R* with genes related to adenosine production, transport, and metabolism in both patient groups (Table 8). The results showed a positive association between *A_2A_R* and *HIF1a* (hypoxia-inducible factor 1-alpha) and a negative correlation with AK (adenylate kinase) in CGGA males. However, in the other group, TCGA astrocytoma, *A_2A_R* was negatively related to *HIF1a, ENPP1* (ectonucleotide pyrophosphatase/phosphodiesterase 1), and pannexin, and positively correlated with *ADK* (adenosine kinase).

## 3. Discussion

Gliomas are the most aggressive brain tumors [5]. Despite the treatment of glioma patients with conventional therapies, the prognosis remains poor. In recent years, immunotherapy has brought new hope as a potential new therapeutic approach for glioma patients. However, the majority of these patients do not respond to the blockade of the usual immune checkpoint pathways (CTLA-4 and PD1/PD-L1) [8], leading to increased interest in investigating other immune checkpoint molecules, including the A_2A_R molecule. A_2A_R has been the subject of several studies, due to its important immunosuppressive effect on different types of cancers, including glioma [20,21,22,23,24,25]. This prompted us to further investigate the expression of A_2A_R at the transcriptomic level, and the immunosuppressive role played by this receptor in the tumor microenvironment of human gliomas.

Our study revealed the following: (1) *A_2A_R* expression level did not correlate with clinical parameters in Moroccan patients. (2) A_2A_R gene expression was related to poor prognosis in Moroccan and Chinese (CGGA) males with gliomas. (3) *A_2A_R* overexpression was related to glioma aggressiveness in CGGA males.(4) *A_2A_R* was related to lymphocyte exhaustion genes and was strongly correlated with genes that are primarily enriched in focal adhesion and extracellular matrix interactions, and that are associated with glioma development and angiogenesis. (5) In the TCGA cohort, *A_2A_R* was related to a good prognosis, and this was observed in patients with an astrocytoma type. (6) The overexpression of *A_2A_R* in TCGA astrocytoma patients was related to pathways reflecting a state of lymphocyte activation. (7) The correlations of *A_2A_R* with adenosine production and metabolism genes suggested that the overexpression of *A_2A_R* in Chinese male patients was associated with an overexpression of extracellular adenosine that binds to *A_2A_R*, to induce an immunosuppressive effect. (8) In the TCGA astrocytoma patients, *A_2A_R* overexpression was associated with extracellular adenosine deficiency and did not show any activation. The lack of activation of this receptor in these patients could protect them from immunosuppression, which may reflect their good prognosis.

We first compared the expression level of *A_2A_R* between subtypes of clinical parameters in Moroccan glioma patients. No statistically significant differences were recorded, suggesting that the expression of this gene was not influenced by clinical parameters. Second, we performed a study of the impact of *A_2A_R* gene expression on the survival of patients with a specific clinical parameter. Our results showed that *A_2A_R* presented a poor prognosis only in Moroccan males. Owing to a limited number of Moroccan patients (only 17 patients or their family members responded to the survival questionnaire), we completed our analysis using another database, the CGGA, which contains a large number of patients. We obtained the same results in CGGA patients, the higher the gene expression, the worse the survival in males. These results suggest the importance of targeting this gene in men, since males with gliomas have a higher incidence and a higher risk of death [26,27,28,29,30], and respond more poorly to treatments [31,32].

To understand the impact of *A_2A_R* on the prognosis of CGGA males with gliomas, we performed a differential analysis followed by KEGG and hallmark enrichment. Our data indicated that *A_2A_R* positively regulated the expression of genes related to focal adhesion signaling pathways and extracellular matrix (ECM) interactions, suggesting that *A_2A_R* may affect the formation of focal adhesions in gliomas and regulate the present ECM genes. Focal adhesion constitutes mechanisms that promote motility, tumor cell invasion, and the migratory and invasive capacity of gliomas. Thus, *A_2A_R* upregulated ECM genes are involved in epithelial mesenchymal transition, angiogenesis, and glioma growth [32,33,34,35]. These results reinforced the role of *A_2A_R* in glioma progression.

However, in the TCGA cohort, the relationship between *A_2A_R* and glioma prognosis differed from the results obtained in the previous two groups. In the TCGA astrocytoma patients, high *A_2A_R* expression was associated with a good prognosis. Enrichment analysis revealed that high *A_2A_R* expression primarily increased with genes involved in the Notch pathway and Th1 and Th2 differentiation. The Notch pathway directly upregulates granzyme B and perforin mRNA expression, promotes differentiation into effector cells, and maintains memory T cells [36,37,38]. In addition, notch deletion results in CD8+ T cell dysfunction and antitumor immunity impairment, whereas stimulation of the notch pathway can increase tumor suppression [36,37,38]. This suggested that CD8+ T cells are functional in the presence of high *A_2A_R* expression in TCGA astrocytoma patients.

T cell exhaustion was identified from a genetic signature of exhausted CD8+ T cells (GET), characterized by the accumulation of multiple co-inhibitory checkpoint molecules and inflammatory genes [39]. In male CCGA patients, the results showed that *A_2A_R* had a positive significant association with six T cell exhaustion genes, which were *LAG-3*, *CTLA-4*, *CXCR6*, *CD276*, *NKG7*, and *HLA-DRB*. However, in the TCGA astrocytoma cohort, there was a negative association of A_2A_R with PD-L1, TIM-3, and HLA-DQA1, and only a positive correlation with *LAG-3*. This was subsequently reinforced by correlations of TCD8 and Treg infiltration levels with cytolytic activity score, which was assessed using granzyme A (*GZMA*) and perforin-1 (*PRF1*), significantly reflecting CD8+ T cell activation and immune status [40,41]. The results showed, in TCGA astrocytoma patients, a positive correlation between CD8+ T cells with cytolytic activity score, suggesting an activated state of the anti-tumor immune response in these patients, whereas, in male CGGA patients, there was no correlation of these cells with cytolytic activity. These results suggest an activated state of lymphocytes, inducing an activated anti-tumor immune response that ensures a good prognosis in TCGA astrocytoma patients with high *A_2A_R* levels.

To understand the causes behind the association of *A_2A_R* with two different prognoses within two populations (CGGA males and TCGA Astrocytoma), we performed a differential analysis based on the *A_2A_R* gene expression in each group. Genes overexpressed in both were excluded. The INOH-path enrichment analysis revealed that the elevated *A_2A_R* expression in CCGA males compared with TCGA astrocytoma was related with upregulation of the pyruvate kinase deficiency, glutaminolysis, hyperphenylalanimia, and sugar metabolism pathways in this group, which, if *A_2A_R* was associated with it, led to a poor prognosis (CGGA men) compared to the TCGA astrocytoma group. Pyruvate kinase is a key enzyme in glycolysis [42]. It can be inhibited by high levels of amino acids [35]. These amino acids, such as phenylalanine and glutamine, are an important source for cancer cells to meet the cell’s demand for ATP under conditions of stress and hypoxia [36]. Therefore, glutaminolysis produces a high concentration of ATP [43,44]. This suggests the high level of ATP led to high adenosine production in the glioma microenvironment in the first group (CGGA men) compared with the second, which would result in a strong immunosuppressive environment.

In addition, because the immunosuppressive effect of *A_2A_R* depends on the amount of extracellular adenosine, we correlated *A_2A_R* with genes related to adenosine production, transport, and metabolism in both conditions. The results revealed a positive association between *A_2A_R* and *HIF1A* and a negative correlation with *AK* in CGGA males. Hypoxia leads to the release of ATP into the extracellular environment [45,46] and plays a key role in establishing an immunosuppressive microenvironment, by increasing the level of extracellular adenosine, which is then bound to *A_2A_R*, to trigger its immunosuppressive effects [16,47,48,49,50]. In addition, adenylate kinase-1 contributes to the regeneration of extracellular ATP [51]. In addition, AK1 knockout mice exhibited an increase in adenosine generation [52]. However, in the other group, TCGA astrocytoma, *A_2A_R* was negatively linked to *HIF1*, *ENPP1*, and pannexin, and positively correlated to *ADK*. Pannexins 1 transmembrane channels are involved in ATP release by inducing an increase in extracellular ATP [53,54,55]. Thus, activation of these channels induces stimulation of A_2A_R via increased extracellular adenosine [56]. Adenosine kinase is an intracellular enzyme that catalyzes the phosphorylation of adenosine to AMP, using ATP as a phosphate donor [57,58,59]. ENPP1 mediates the conversion of ATP to AMP, which is then converted to adenosin [51,60,61]. All in all, these results suggest that the overexpression of *A_2A_R* in Chinese male patients was associated with overexpression of extracellular adenosine, which binds to *A_2A_R*, to induce an immunosuppressive effect and consequently a poor prognosis in these patients. However, in the second group (TCGA astrocytomas), *A_2A_R* overexpression was associated with adenosine deficiency suggesting the absence of activation. The absence of activation of this receptor in TCGA astrocytoma patients could have protected them from immunosuppression, which translated into their good prognosis.

## 4. Materials and Methods

### 4.1. Patients and Samples

mRNA expression was assessed in a total of 52 specimens from glioma patients that were collected at the Ibn Rochd University Hospital, Department of Neurosurgery (Casablanca, Morocco). The following inclusion criteria were agreed upon: Informed consent to participate in the study protocol, full documentation of the study, and patients diagnosed with glioma. However, exclusion criteria consisted of incomplete study documentation and no informed consent available to participate in the study protocol. Samples were recruited from 2016 to 2019. All glioma tissues were classified according to the World Health Organization (WHO) 2007 and 2016 [62,63]. Clinical information was obtained from patients’ medical reports.

### 4.2. Total RNA Isolation and Reverse Transcription (RT)

Total RNA was extracted from glioma specimens using TRIzol reagent (Invitrogen, Massy, France), as detailed previously [64,65,66]. RNA concentration was determined using a NanoVueTM Plus spectrophotometer (GE Healthcare, Hatfield, UK). We then diluted the samples with ultrapure water, in order to have same concentration of RNA per tube. In accordance with the instructions of the manufacturer, cDNA was initially synthesized using the Tetro reverse transcriptase enzyme (Bioline, Livron-sur-Drôme, France) from 0.5 μg of total RNA in a 20 μL reaction mixture with 1 μL of 25 µg random hexamer primer (Bioline, France) and 4 μL of RNase-free water was added and incubated at 70 °C for 5 min. After that, 4 μL of dNTP (10 mM), 4 μL of Tetro reverse transcriptase buffer, 0.5 μL of Tetro reverse transcriptase enzyme (Bioline, France), 0.5 μL of RNase inhibitor (Invitrogen, France), and 1 μL of RNase-free water were added and then incubated at 25 °C for 10 min, followed by 45 °C for 30 min, and finally 85 °C for 5 min.

### 4.3. RT-PCR (Real-Time—Polymerase Chain Reaction)

During the RT-PCR, a quantitative analysis of the *A_2A_R* gene expression was performed using “Applied Biosystems™ 7500 Fast software v2.0.6”. In the PCR plate wells, a reagent mixture (18 µL) was introduced: Ultra-pure water (7 µL), primers “Forward” (0.5 µL) and “Reverse” (0.5 µL), SYBR Green (10 µL), and 2 µL of cDNA per well. The NTC (no template control), negative control contained 2 µL of ultra-pure water instead of cDNA. The positive control represented the sample already expressing the β-actin or A_2A_R gene and was used to validate the overall workflow of the manipulation. The PCR plate was placed in the thermal cycler following 40 cycles of 10 min at 95 °C, 15 s at 95 °C, and 1 min at 60 °C. The results were interpreted by exploiting the threshold cycle and melt curve. The results were analyzed in relation to Beta-actin expression using the 2^ ^(−ΔCT)^ method described by Livak and Schmittgen [67].

The primers of *A_2A_R* and β-actine genes:β-actine: Forward: 5′ TGGAATCCTGTGGCATCCATGAAAC-3′β-actine: Reverse: 5′-TAAAACGCAGCTCAGTAACAGTCCG-3′*A_2A_R*: Forward:5′ ATC GCC ATT GAC CGC TAC AT3′*A_2A_R*: Reverse: 5′ GCT GAC CGC AGT TGT TCC A3′

### 4.4. TCGA and CGGA Clinical and Transcriptomic Data Collection

Molecular data and clinical data, including primary glioma mRNA expression of CGGA(n = 229), LGG (n = 516) and GBM (n = 154) patients, were downloaded from The Cancer Genome Atlas (TCGA) [68] data portal (https://www.cbioportal.org/ (accessed on 22 April 2021)) and CGGA_325 (http://www.cgga.org.cn/ (accessed on 30 April 2021)) [69].

### 4.5. Differential Analyses

Differential analyses of genes expressed in the tumor microenvironment were performed using the R limma package [70]. Lowly expressed genes were removed to correct for the batch effect. The calcNormFactors function was used to calculate the normalization factor for each patient, and the voom function of the limma package was used to perform CPM normalization, adjusted by the TMM method. RNA-seq values were normalized by quantile. Differentially expressed genes (DEGs) were filtered by applying a false discovery rate (FDR) of less than 0.05 to the adjusted P values, which were generated using the approach of Benjamini and Hochberg [71]. The results were visualized in the form of volcanoplots and tables.

### 4.6. Enrichment Analyses

To determine the networks, functional analyses, and pathways that DEGs might involve, we performed enrichment, which was done using the gene list and the resulting FC (fold change) from the differential analysis. Terms were considered significantly enriched if the log2Fold change was greater or equal 0.5 and *p* value < 0. The databases used were KEGG (Kyoto Encyclopedia of Genes and Genomes) and INOH-path (The Integrating Network Objects with Hierarchies). The package used was pathfindeR [72] by R programming and also the ShinyGO website http://bioinformatics.sdstate.edu/go74/ (accessed on 24 December 2021).

### 4.7. Analysis of Intra-Tumor Immune Cell Composition

To quantify the level of immune cell infiltration in the glioma samples based on gene expression, we used CIBERSORTx “Cell type Identification By Estimating Relative Subsets Of RNA Transcripts”, which is a deconvolution algorithm developed by Newman et al. [73,74] to characterize the composition of immune cells in tissues based on a gene signature array, called LM22 [73,74]. The algorithm was run using the LM22 signature, enabling batch correction in B mode, and the setting of permutations for statistical analysis. The permutation chosen was 100. (The 1000 permutation showed the same results).

### 4.8. Statistical Analysis, Survival, and Graph Generation

To compare the gene expression between groups, a parametric *t*-test was performed for groups above 30 and a Mann–Whitney test for groups below. We also employed a Kruskal–Wallis test to compare the expression of different genes in the same group of patients. An ANOVA test was used to compare three groups. For the analysis of the correlation between the expression of two different genes in the same group, a non-parametric Spearman test was performed. Kaplan–Meier survival (Log-Rank test) analysis was performed to compare the survival between different groups. Survival was also analyzed with a (Mantel–Cox) multivariate test. All statistical analyses and plots were performed using GraphPad Prism 6.0 (GraphPad Software, Inc., La Jolla, CA, USA) and RStudio 1.4.1717 (https://www.rstudio.com/products/rstudio/download/ (accessed on 27 May 2021)). HeatMaps were generated using the Morpheus platform (https://software.broadinstitute.org/morpheus). Cytolytic activity score was calculated as the geometric mean of *GZMA* and *PRF1* (CYT score = √ *GZMA* × *PRF1*) for each patient [75,76]. Differences with *p* ≤ 0.05 were considered statistically significant.

## Figures and Tables

**Figure 1 ijms-24-06688-f001:**
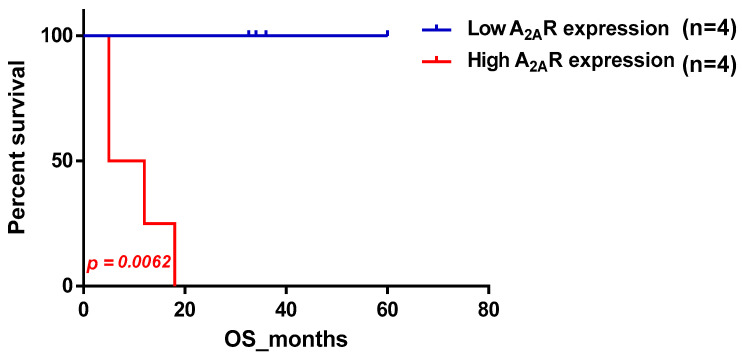
Survival curve according to *A_2A_R* expression, indicating poor survival in Moroccan males. Patients were divided into two groups based on median *A_2A_R* expression. Blue curve represents patients with low expression and red represents high *A_2A_R* expression.

**Figure 2 ijms-24-06688-f002:**
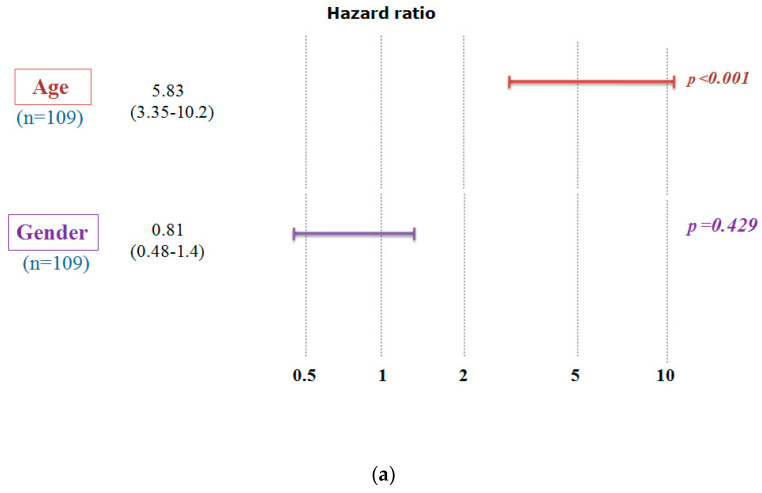
Multivariate Cox analysis demonstrates that the poor survival of CGGA patients according to *A_2A_R* expression level is influenced only by gender. A hazard ratio (HR) value greater than 1 indicates that high gene expression gives a poor prognosis. (**a**) Low *A_2A_R* expression and (**b**) High *A_2A_R* expression. (Codes: Age < 44 years: 0; Age > 44 years: 1; Women: 0; Men: 1).

**Figure 3 ijms-24-06688-f003:**
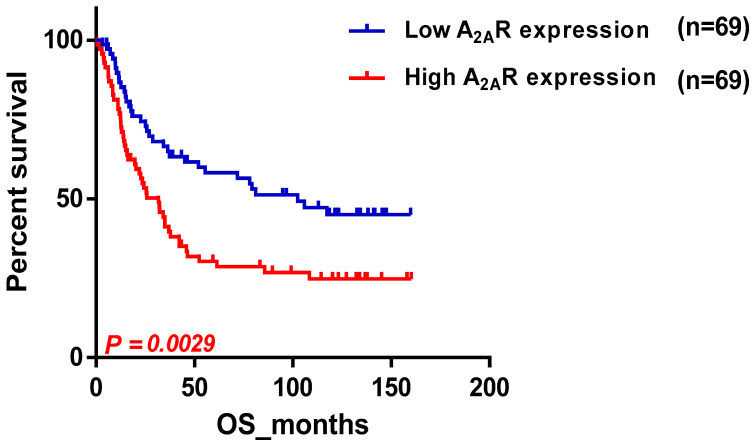
Survival curve according to *A_2A_R* expression, indicating poor survival in CGGA males. Patients were divided into two groups based on median *A_2A_R* expression. Blue curve represents patients with low expression, and red curve represents high *A_2A_R* expression.

**Figure 4 ijms-24-06688-f004:**
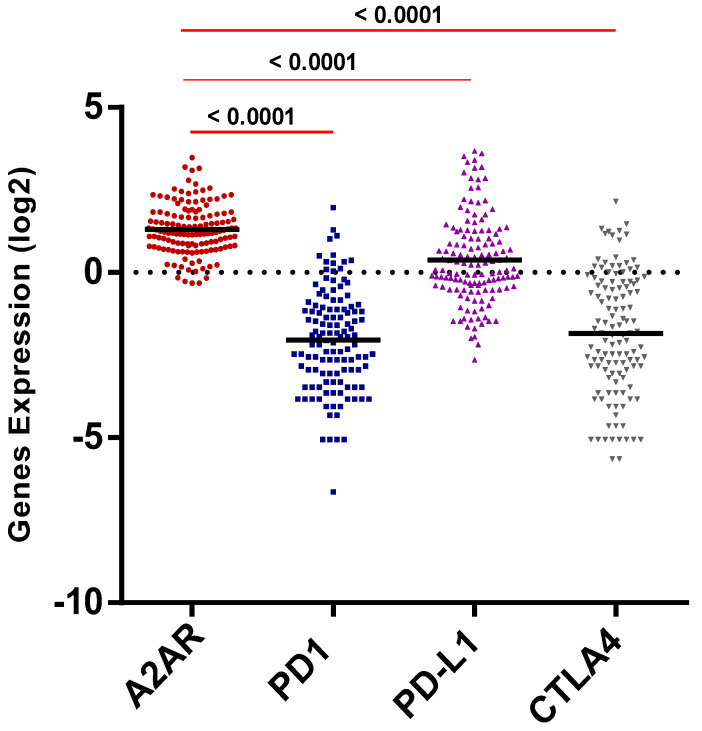
*A_2A_R* expression level was higher than PD1, PD-L1, and CTLA4.

**Figure 5 ijms-24-06688-f005:**
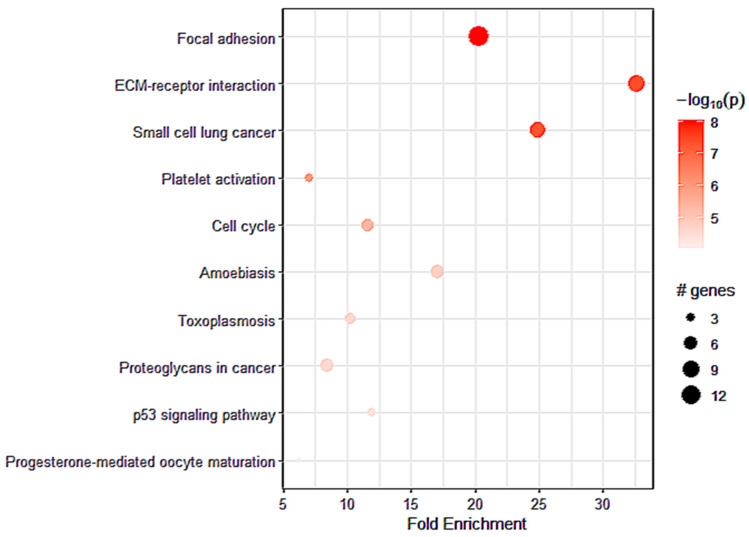
The top pathways for strong *A_2A_R* expression obtained by KEGG enrichment in CGGA males.

**Figure 6 ijms-24-06688-f006:**
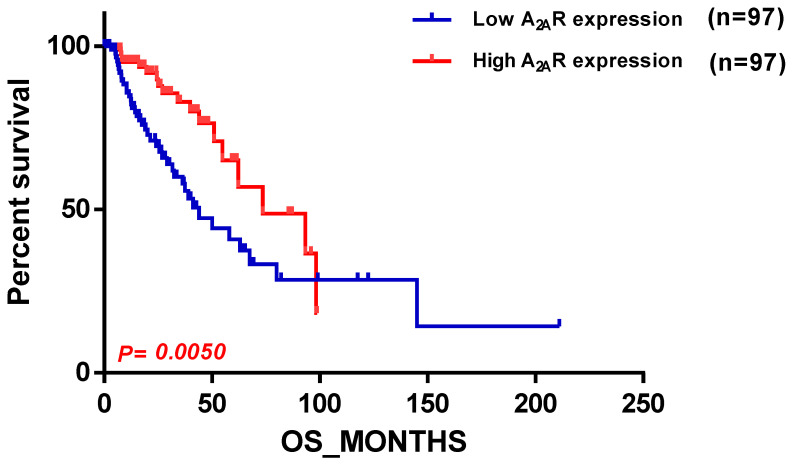
Survival curve according to *A_2A_R* expression, indicating good prognosis survival in TCGA astrocytoma. Patients were divided into two groups based on median *A_2A_R* expression. Blue curve represents patients with low expression, and red curve represents high *A_2A_R* expression.

**Figure 7 ijms-24-06688-f007:**
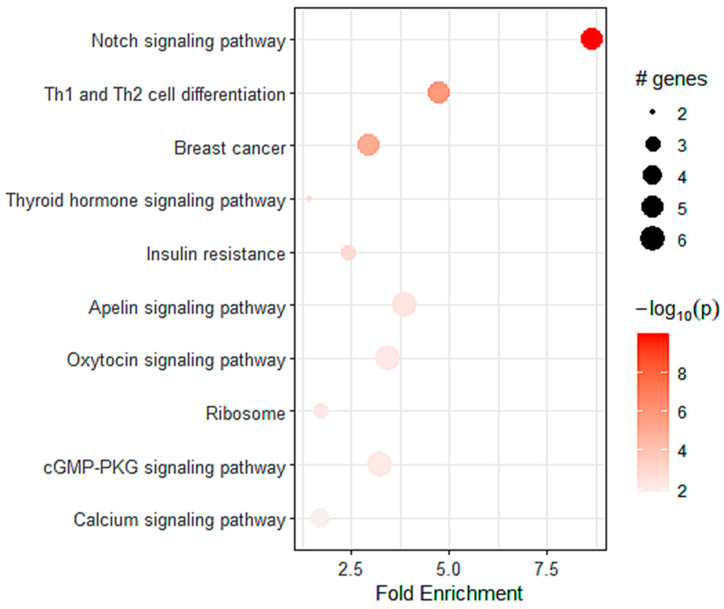
The top pathways for strong *A_2A_R* expression obtained by KEGG enrichment in TCGA astrocytoma patients.

**Figure 8 ijms-24-06688-f008:**
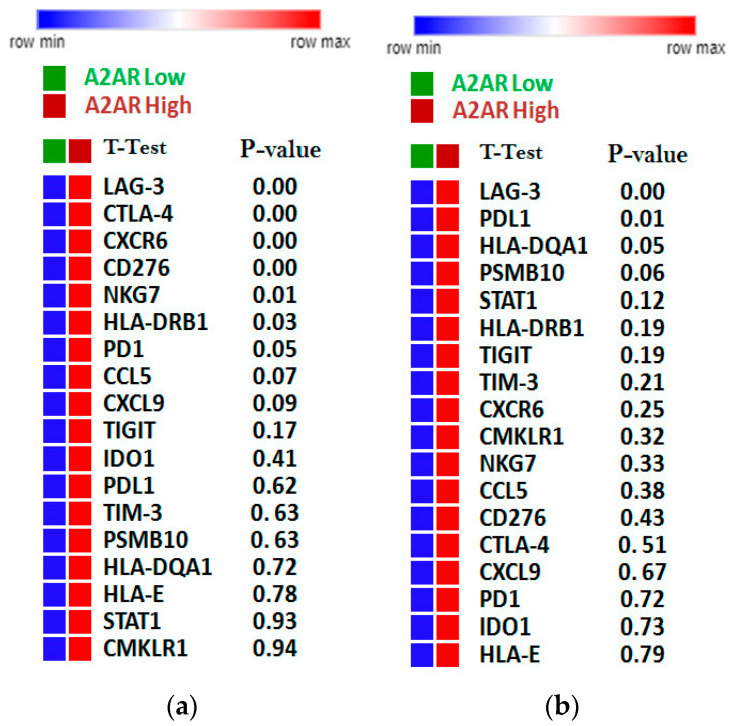
Correlation of *A_2A_R* expression with lymphocyte exhaustion genes. (**a**) The CGGA male cohort; (**b**) The TGGA astrocytoma cohort. The more intense the red color, the stronger the gene expression; Green/orange color: Low A_2A_R/High *A_2A_R*.

**Figure 9 ijms-24-06688-f009:**
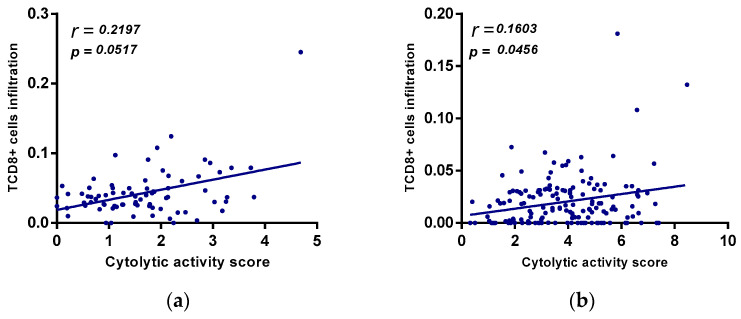
(**a**) positive correlation between *A_2A_R* expression and cytolytic activity score in TCGA astrocytoma. Correlation between cytolytic activity score and TCD8+ cells in CGGA males. (**b**) Correlation of cytolytic activity score and TCD8+ cells in TGGA astrocytoma.

**Figure 10 ijms-24-06688-f010:**
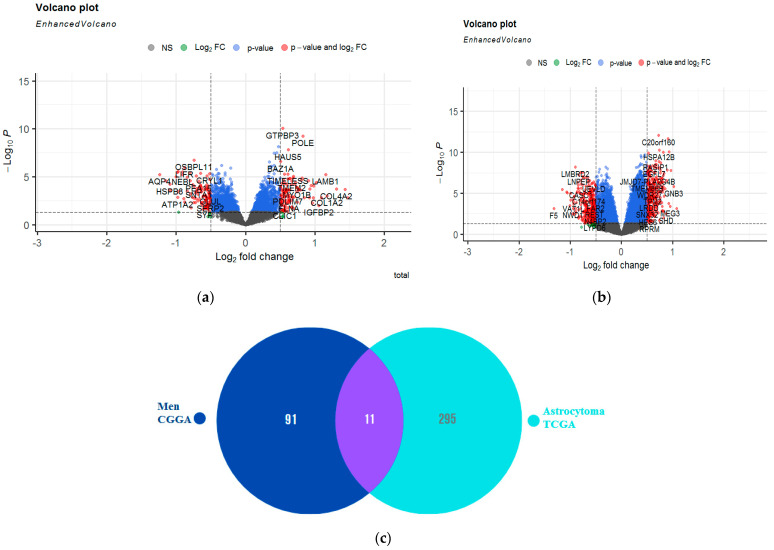
Identification of genes that were uniquely overexpressed in the CCGA males compared with the TCGA astrocytoma group according to *A_2A_R* expression. (**a**) Volcano plot showing genes differentially expressed according to *A_2A_R* expression in CGGA male patients (right of volcano) (fold change ≥ 0.5; *p* value ≤ 0.05). (**b**) Volcano plot showing genes differentially expressed according to *A_2A_R* expression in TGGA astrocytoma patients (right of volcano) (fold change ≥ 0.5; *p* value ≤ 0.05); (**c**) Venn diagram showing genes not in common between the two groups.

**Table 1 ijms-24-06688-t001:** *A_2A_R* gene expression was not correlated with clinical parameters in Moroccan patients.

Variable	Case(%)	*p*-Value
	(n = 52)	
Gender		
•Male	31 (59.62)	0.0963
•Female	21 (40.38)	
Age		
•≤34 years	25 (52.083)	0.3754
•˃34 years	23 (47.916)	
WHO Grade		
•I	17 (33)	0.5047
•II	9 (17)	
•III	6 (12)	
•IV	20 (38)	
Histological Type		
•Astrocytomas	19 (38.775)	0.3109
•Oligodendrogliomas	3 (6.122)	
•Ependymomas	7 (14.285)	
•Glioblastoma	20 (40.816)	
Smoking status		
•Yes	11 (26,19)	0.1985
•No	31 (73,80)	
IDH mutation		
•WT	11 (73,33)	0.7824
•IDH mutant	4 (26,66)	

**Table 2 ijms-24-06688-t002:** *A_2A_R* gene expression was associated with worse survival in Moroccan male patients.

Variable	Cas (Low/High)	*p*-Value
Overall survival	9/8	0.8232
Gender		
•Male	4/4	0.0062
•Female	5/4	0.0532
Age		
•≤34 years	4/4	0.8629
•˃34 years	4/4	0.7297
WHO Grade		
•LGG	4/4	0.3173
•GBM	5/4	0.8781

**Table 3 ijms-24-06688-t003:** Impact of *A_2A_R* gene expression level on poor prognosis of CGGA patients with sub-median age or males.

	Nb Low/High	*p*-Value
Age		
≤44 years	62/58	0.0239
˃44 years	51/51	0.7516
Gender		
Male	69/69	0.0029
Female	43/41	0.5453
WHO Grade		
GII	45/45	0.4268
GIII	24/23	0.5753
GIV or GBM	43/42	0.8806
Histological type		
Oligodendroglioma	22/22	0.704
Astrocytoma	23/23	0.887
IDH mutation		
IDH mutant	57/37	0.1249
WT	55/54	0.6984

**Table 4 ijms-24-06688-t004:** The top pathways for strong *A_2A_R* expression obtained by Hallmark enrichment in CGGA males.

Enrichment FDR	Hallmark Enrichment
1.3 × 10^−20^	Hallmark epithelial mesenchymal transition
2 × 10^−4^	Hallmark TGF beta signaling
1.6 × 10^−2^	Hallmark angiogenesis
1.4 × 10^−6^	Hallmark apoptosis
7.1 × 10^−7^	Hallmark G2M checkpoint

**Table 5 ijms-24-06688-t005:** *A_2A_R* gene expression was associated with good prognosis in TCGA astrocytoma patients.

	Nb Low/High	*p* Value
Age		
≤44 years	157/157	0.6498
˃44 years	152/152	0.1307
Gender		
•Male	175/173	0.1577
•Female	135/135	0.6536
Grade		
GII	124/124	0.2578
GIII	132/132	0.5293
GIV or GBM	75/73	0.4233
Histological type		
Oligodendroglioma	95/94	0.1064
Oligoastrocytoma	65/65	0.684
Astrocytoma	97/97	0.005
IDH status		
IDH mutant	212/210	0.958
WT	99/98	0.248

**Table 6 ijms-24-06688-t006:** The top pathways for strong *A_2A_R* expression obtained by Hallmark enrichment in TCGA astrocytoma patients.

Enrichment FDR	Hallmark Enrichment
4.2 × 10^−3^	Hallmark myogenesis

**Table 7 ijms-24-06688-t007:** Principal pathways overexpressed in CGGA males compared with the TCGA astrocytoma group in correlation with the *A_2A_R* expression obtained through INOH enrichment.

Enrichment FDR	Pathways Involved
2.385 × 10^−7^	Pyruvate kinase deficiency
0.01043	Glutaminolysis and cancer
0.01043	Hyperphenylalanemia
0.01043	Ibutilide action pathway
0.01043	Metabolism of nucleotide sugars

**Table 8 ijms-24-06688-t008:** Association of *A_2A_R* expression with genes related to adenosine production and metabolism, in CGGA males and TCGA astrocytoma, suggesting elevated extracellular adenosine levels in CGGA males.

	Males CGGA		Astrocytoma TGGA	
	*p*	r	*p*	r
*HIF1A*	0.0452	0.1684	0.0005	−0.2464
*CD39*	0.3698	0.07582	0.9703	0.002686
*NT5E*	0.2907	0.08927	0.3233	0.07128
*ENPP1*	0.5992	0.04513	<0.0001	−0.2757
*CD38*	0.0095	−0.217	0.0234	−0.1627
*BST1*	0.9523	0.005137	0.7136	0.02652
*AK1*	0.0465	−0.1698	0.8975	−0.009307
*ADA*	0.0001	0.3198	0.0468	0.1429
*PNP*	0.7196	−0.03083	0.5284	−0.04554
*ADK*	0.9816	−0.001977	0.0483	0.142
Connexin	0.0059	−0.2299	0.018	−0.1698
Pannexin	0.0782	0.1483	<0.0001	−0.2851

## Data Availability

Not applicable.

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
