# Peer review of "A_2A_R as a Prognostic Marker and a Potential Immunotherapy Target in Human Glioma"

_ijms, 2023, doi:10.3390/ijms24076688_

Round 1

Reviewer 1 Report

The authors aim to analyze the role of adenosine 2A receptor (A2AR) overexpression in glioma patients. By using qRT-PCR in Moroccan, TCGA, and CGGA primary glioma patients, they mentioned that high expression of this gene was associated with poor survival in men. They also mentioned that the A2AR gene expression level played an important role in the tumor microenvironment. I have several comments for the authors’ consideration to further improve the manuscript.

1.      A2AR mRNA expression in glioma patients from 52 patients with glioma was subjected to RT-PCR. The limitation of this part was that of a small sample size. The sample size was too small to draw these conclusions, especially in the survival analysis which only included 17 patients who had the survival data. It is highly recommended that the authors could explain this issue in the discussion section.

2.      If possible, I also suggest the author run an IHC analysis of A2AR to further support their results.

3.      Of all these cases (patient number) in the overall survival part were 17 patients In Table 2. However, in the gender and age part, there were only 16 patients, the author needs to recheck the original data.

4.      In 2.3. the author mentioned that A2AR expression is most elevated compared to PD-1, PD-L1, and CTLA-4 in male CGGA. However, I don’t think this analysis could draw the conclusion that A2AR play an important role in the tumor microenvironment. The author needs to do the correlation analysis between A2AR expression and the expression of these immune checkpoints, instead of comparing the expression of these genes.

5.      In the enrichment analysis the author only used the KEGG enrichment analysis, I also suggest the author could use the Hallmark gene set, which was more cancer-specific (Both in TCGA and CGGA).

6.      Figure 8. Is not clear enough, please use a higher-resolution figure.

7.      Figure 9. is blank, please check.

8.      There are some minor language errors. The authors should be revised the manuscript with an English language editor to make it more readable.

Author Response

Point by point response to reviewers' comments

We thank the reviewers for their interest in our study, namely “A2AR as a prognostic marker and a potential immunotherapy target in human glioma”.

We report below a point-by-point response to all comments:

Reviewer 1

  • Comment number 1:

A2AR mRNA expression in glioma patients from 52 patients with glioma was subjected to RT-PCR. The limitation of this part was that of a small sample size. The sample size was too small to draw these conclusions, especially in the survival analysis which only included 17 patients who had the survival data. It is highly recommended that the authors could explain this issue in the discussion section.

Response to comment number1:

Thank you for your comment. We were not able to get back fromall the patients who participated to the study, only 17 out of the 52 patients responded to our request. Thishas now been discussedin the new version of the manuscript.

  • Comment number 2:

If possible, I also suggest the author run an IHC analysis of A2AR to further support their results.

Response to comment number 2:

We agree with the reviewer suggestion, andthe expression analysis of the A2aR by immunohistochemistry was indeed planned, but due to constraints in terms of time and receipt of products, we have programmed this protein study for the next paper. On the other hand, we believe that our data are overall consistent and compatible, given the three independent cohorts we have used.

  • Comment number 3:

Of all these cases (patient number) in the overall survival part were 17 patients In Table 2. However, in the gender and age part, there were only 16 patients, the author needs to recheck the original data.

Response to comment number 3:

The error has been rectified. Thank you for noticing this. For gender, there are in fact 17 patients, of which 9 are female and 8 are male. However, for age, we only have age data for 16 patients. As the age of the 17th patient is unknown, we did not include him in the survival study for age.

Comment number 4:

In 2.3. the author mentioned that A2AR expression is most elevated compared to PD-1, PD-L1, and CTLA-4 in male CGGA. However, I don’t think this analysis could draw the conclusion that A2AR play an important role in the tumor microenvironment. The author needs to do the correlation analysis between A2AR expression and the expression of these immune checkpoints, instead of comparing the expression of these genes.

Response to comment number 4:

We agree with this remark. We have now removed that statement in the new version of the manuscript. On the other hand, regarding correlations, we have compared the expression level of PD1, CTLA and PDL1 between patients with low A2AR and patients with high A2AR in figure 8a. As a result, A2AR and CTLA-4 are positively and significantly correlated, whereas PD1, PDL1, and A2AR are not.

  • Comment number 5:

In the enrichment analysis the author only used the KEGG enrichment analysis, I also suggest the author could use the Hallmark gene set, which was more cancer-specific (Both in TCGA and CGGA).

Response to comment number 5:

Thank you for this remark, the Hallmark enrichment has now been added and described in the new version of the manuscript.

  • Comment number 6:

Figure 8 is not clear enough, please use a higher-resolution figure.

Response to comment number 6:

Thank you for noticing this important point. The quality of the figure has now been improved; and the new figure seems now much better in the new version of the manuscript.

  • Comment number 7:

Figure 9. is blank, please check.

Response to comment number 7:

We totally agree with the remark. The figure has been reinserted in the new version of the manuscript.

  • Comment number 8:

There are some minor language errors. The authors should be revised the manuscript with an English language editor to make it more readable.

Response to comment number 8:

Thank you for noticing this point. The language has been reviewed by an English-speaking person, and has now been improved in the new version of the manuscript.

Reviewer 2 Report

The work is interesting and well conducted, but some improvement in terms of readability should be made. Here several observations

·         Authors are encouraged to include these recent reviews on A2 receptor PMID: 28174424, PMID: 31189400

·         I suggest reducing the length of the abstract in order to improve the reading, same observation for the discussion

·         Introduction, among adenosine receptors, besides A2AR, A3 receptor has been frequently reported as overexpressed in Glioblastoma https://www.nature.com/articles/s41388-021-02090-z please include this notion and supporting reference

·         The quality of figure 8 should be improved

·         Figure 9 is unreadable

·         Line 32 “and a successful” please revise the typo

·         Line 87 better “low median expression” and “high median expression”

·         Please include the exact number of patients in each study cohort in the captions of all figures with survival curves

·         Authors are encouraged to uniform the size of the two vulcano plots depicted in figure 10

·         Were samples negative for A2aR expression? And if yes, where these samples included among low A2Ar expression for survival analyses?

·         Line 103 “(Table3).” please revise the typo. I have noted the same error in other sections of the manuscript, e.g., line 186, line 226 etc.., when tables are mentioned please revise it

·         Lines 99 and 125 better patients instead of men. If the authors are referring to the gender, the correct annotation is males and females. For instance in line 273, are the authors meaning males?

·         Line 168 A sentence should not start with a number

·         Line 180 numerosity?

·         Line 242 a reference should be included

·         Please include supporting references in the methods

·         Were the primers for A2AR specific for all gene isoforms? https://www.ensembl.org/Homo_sapiens/Gene/Summary?db=core;g=ENSG00000128271;r=22:24417879-24442357

Author Response

Point by point response to reviewers' comments

We thank the reviewers for their interest in our study, namely “A2AR as a prognostic marker and a potential immunotherapy target in human glioma”.

We report below a point-by-point response to all comments:

Reviewer 2

  • Comment number 1:

 Authors are encouraged to include these recent reviews on A2 receptor PMID: 28174424, PMID: 31189400

Response to comment number 1:         

Thank you for this important remark. Recent reviews on A2 receptor have now been added in the new version of the manuscript.

  • Comment number 2:

I suggest reducing the length of the abstract in order to improve the reading, same observation for the discussion

Response to comment number 2:

We agree with this remark. The length of the abstract and the discussion has now been reduced in the new version of the manuscript.

  • Comment number 3:

Introduction, among adenosine receptors, besides A2AR, A3 receptor has been frequently reported as overexpressed in Glioblastoma https://www.nature.com/articles/s41388-021-02090-z please include this notion and supporting reference

    Response to comment number 3:

We agree with the remark. “The notion: Introduction, among adenosine receptors, besides A2AR, A3 receptor has been frequently reported as overexpressed in Glioblastoma”, has now been inserted in the new version of the manuscript. The indicated reference has been inserted as well.

  • Comment number 4:

The quality of figure 8 should be improved

Response to comment number 4:

The figure has now been improved in the new version of the manuscript

  • Comment number5:

Figure 9 is unreadable

    Response to comment number 5:

The quality of the figure has now been improved and reinserted in the new version of the manuscript.

Comment number 6:

Line 32 “and a successful” please revise the typo

Response to comment number 6:

The spelling mistake has now been revised in the new version of the manuscript.

  • Comment number 7:

Line 87 better “low median expression” and “high median expression”

Response to comment number 7:

We totally agree with suggestion, which has now been modified in the new version of manuscript.

  • Comment number 8:

Please include the exact number of patients in each study cohort in the captions of all figures with survival curves

Response to comment number 8:

The exact number of patients in each study cohort has now been added to all figures with survival curve in the new version of the manuscript 

  • Comment number 9:

Authors are encouraged to uniform the size of the two volcano plots depicted in figure 10

Response to comment number 9:

Thank you for noticing this important point. The size of the two volcano plots in figure 10 has now been standardized in the new version of  the manuscript.

  • Comment number 10:

Were samples negative for A2aR expression? And if yes, where these samples included among low A2AR expression for survival analyses?

Response to comment number 10:

In our study, there were no samples negative for A2aR expression.

  • Comment number 11:

Line 103 “(Table3).” please revise the typo. I have noted the same error in other sections of the manuscript, e.g., line 186, line 226 etc.., when tables are mentioned please revise it

Response to comment number 11:

The spelling mistake has now been revised in the new version of the manuscript.

  • Comment number 12:

Lines 99 and 125 better patients instead of men. If the authors are referring to the gender, the correct annotation is males and females. For instance in line 273, are the authors meaning males?

Response to comment number 12:

Thank you for noticing this important point..We have now replaced the term “men” by “males” in the new version of the manuscript. 

  • Comment number 13:

Line 168 A sentence should not start with a number

Response to comment number 13:

The line 168 has now been revised in the new version of the manuscript.

  • Comment number 14:

Line 180 numerosity?

Response to comment number 14:

The spelling mistake has nowbeen revised in the new version of the manuscript. .

  • Comment number 15:

Line 242 a reference should be included.

Response to comment number 15:

The reference in line 242 has now been included in the new version of the manuscript.

  • Comment number 16:

Please include supporting references in the methods

Response to comment number 16:

We agree with the remark. Supporting references have now been included in the new version of the manuscript.

  • Comment number 17:

Were the primers for A2AR specific for all gene isoforms?

 https://www.ensembl.org/Homo_sapiens/Gene/Summary?db=core;g=ENSG00000128271;r=22:24417879-24442357

Response to comment number 17:

Primers were specific to all A2AR human gene.

Below is the link to the exons containing the primer sequences:

https://www.ensembl.org/Homo_sapiens/Transcript/Exons?db=core;g=ENSG00000128271;r=22:24417879-24442357;t=ENST00000337539

Round 2

Reviewer 1 Report

The authors addressed most of my concerns; now the paper is more readable.